# Clinical Characteristics, Diagnosis, and Management of Primary Malignant Lung Tumors in Children: A Single-Center Analysis

**DOI:** 10.3390/biomedicines13081824

**Published:** 2025-07-25

**Authors:** Mihail Basa, Nemanja Mitrovic, Dragana Aleksic, Gordana Samardzija, Mila Stajevic, Ivan Dizdarevic, Marija Dencic Fekete, Tijana Grba, Aleksandar Sovtic

**Affiliations:** 1Department of Pulmonology, Mother and Child Health Care Institute of Serbia, 11070 Beograd, Serbia; mihail.basa@imd.org.rs (M.B.); tijana.grba@imd.org.rs (T.G.); 2Pathology Department, Mother and Child Health Care Institute of Serbia, 11070 Beograd, Serbia; nemanja.mitrovic@imd.org.rs (N.M.); gordana.samardzija@imd.org.rs (G.S.); 3Department of Hemato-Oncology, Mother and Child Health Care Institute of Serbia, 11070 Beograd, Serbia; dragana.aleksic@imd.org.rs; 4Cardiac Surgery Department, Mother and Child Health Care Institute of Serbia, 11070 Beograd, Serbia; mila.stajevic@imd.org.rs (M.S.); ivan.dizdarevic@imd.org.rs (I.D.); 5Faculty of Medicine, University of Belgrade, 11000 Belgrade, Serbia; marijadfekete@yahoo.com; 6Institute for Pathology, 11000 Belgrade, Serbia

**Keywords:** pediatric lung tumors, bronchoscopy, chest CT, surgical resection, fusion genes, targeted therapy

## Abstract

**Background/Objectives**: Primary malignant lung tumors in children are rare and diagnostically challenging. This study presents a single-center experience in the diagnosis and treatment of these tumors, emphasizing the role of histopathological and genetic profiling in informing individualized therapeutic strategies. **Methods**: We retrospectively reviewed records of seven pediatric patients (ages 2–18) treated from 2015 to 2025. Diagnostics included laboratory tests, chest CT, bronchoscopy, and histopathological/immunohistochemical analysis. Treatment primarily involved surgical resection, complemented by chemo-, radio-, or targeted therapies when indicated. **Results**: Inflammatory myofibroblastic tumor (IMT) represented the most commonly diagnosed entity (3/7 cases). The tumors presented with nonspecific symptoms, most frequently dry cough. Tumor type distribution was age-dependent, with aggressive forms such as pleuropulmonary blastoma predominantly affecting younger children, whereas IMT and carcinoid tumors were more common in older patients. Surgical resection remained the mainstay of treatment in the majority of cases. Bronchoscopy served as a valuable adjunct in the initial management of tumors exhibiting intraluminal growth, allowing for direct visualization, tissue sampling, and partial debulking to alleviate airway obstruction. In patients with an initially unresectable IMT harboring specific gene fusion rearrangement (e.g., TFG::ROS1), neoadjuvant targeted therapy with crizotinib enabled adequate tumor shrinkage to allow for subsequent surgical resection. Two patients in the study cohort died as a result of disease progression. **Conclusions**: A multidisciplinary diagnostic approach—integrating radiologic, bronchoscopic, histopathological, and genetic evaluations—ensures high diagnostic accuracy. While conventional treatments remain curative in many cases, targeted therapies directed at specific molecular alterations may offer essential therapeutic options for selected patients.

## 1. Introduction

Primary lung tumors in children are a group of rare and mostly benign lesions that differ in etiology and radiologic, bronchologic, and pathologic features [1]. Primary malignant tumors of the lung in the pediatric population are exceptionally rare, comprising less than 1% of childhood cancers [1,2]. Despite their rarity, these neoplasms may arise from virtually any anatomical structure within the tracheobronchial tree, including the epithelial cells, vascular endothelium, lymphatic channels, neural components, and pleural tissue. Nevertheless, certain tumor types are observed more frequently, with age playing an important role in their distribution [1,2]. In early childhood, particularly among the youngest patients, pleuropulmonary blastoma and infantile fibrosarcoma are more commonly reported, whereas in older children, inflammatory myofibroblastic tumors and carcinoid tumors are predominant. These malignancies differ considerably in their growth dynamics and biological aggressiveness—from indolent tumors such as typical carcinoids to rapidly proliferating and poorly differentiated entities like NUT carcinoma and small cell carcinoma [1,2,3]. Prognosis varies accordingly, depending on the histologic subtype, rate of progression, and stage at diagnosis, underscoring the need for tailored therapeutic strategies and vigilant clinical management [1,2].

Since patients with malignant lesions usually present with heterogeneous and nonspecific clinical signs and symptoms, diagnostic efforts focus on early recognition and prompt treatment [1]. The low incidence of these tumors, the heterogeneity of local protocols, and limited access to advanced healthcare remain among the major challenges faced by the healthcare systems. The lack of international studies on standards of treatment often impedes the adoption of uniform recommendations about diagnosis, treatment, and follow-up of those diagnosed with malignant lung masses during childhood.

The inability to establish a definitive diagnosis based solely on clinical presentation and chest X-ray findings has led to increased reliance on more advanced diagnostic modalities [4]. While chest CT frequently reveals key distinguishing features that contribute to an accurate diagnosis, overlapping imaging patterns often necessitate adjunctive tools capable of providing precise interpretation [5]. Although open-lung biopsy remains the gold standard, the ongoing development and clinical implementation of novel, high-yield bronchoscopic techniques have enabled a less invasive yet highly reliable approach to histopathologic assessment. Currently, flexible bronchoscopy—augmented by advanced techniques such as endobronchial ultrasound–transbronchial needle aspiration (EBUS-TBNA), cryobiopsy, laser, and cautery—has become an indispensable diagnostic tool with growing applicability in the treatment of malignant lesions in selected cases [6,7,8,9]. In parallel, ongoing progress in understanding the molecular mechanisms driving tumor proliferation has increasingly facilitated the incorporation of targeted therapies into clinical practice. These agents are designed to inhibit specific signaling pathways essential for tumor cell growth and survival, and offer promising potential for unresectable or disseminated lesions [10,11,12]. Accordingly, the molecular analysis of tumor tissue, including the detection of chromosomal rearrangements and genetic variations, has become a key component of contemporary diagnostic protocols [10,11,12].

Nonetheless, the management of pediatric patients with pulmonary malignancies remains particularly challenging in resource-limited settings with restricted access to advanced diagnostic and therapeutic modalities. Given the limited number of studies addressing pediatric pulmonary malignancies outside high-income countries, the primary objective of this study is to illustrate a pragmatic diagnostic and therapeutic approach tailored to children with primary lung tumors in under-resourced healthcare systems.

A secondary aim is to emphasize the importance of postoperative histopathologic evaluation and chromosomal rearrangement testing using specific biomarkers. Due to the high mitotic index and associated genomic instability of tumor tissue, the emergence of previously undetected chromosomal or genetic abnormalities is possible. Within this context, particular attention is devoted to a case involving a child diagnosed with type II pleuropulmonary blastoma and a previously unreported genetic variant.

The parents provided written consent for the publication of the data. The local ethics committee approved the conduct and publication of this study (decision number 11140/1 and date of approval: 26 December 2024).

## 2. Materials and Methods

This retrospective study included medical records of the patients diagnosed with primary lung tumors at the Mother and Child Health Institute of Serbia, a national tertiary care university hospital, from 2015 to 2025. The analyzed data included patients’ demographics (age and sex), clinical presentation, and findings from the diagnostic workup (laboratory tests, imaging, and bronchoscopy), as well as histopathologic reports. Data on treatment, disease complications, and clinical outcomes were also collected. Children were excluded if they had a diagnosis of primary extrapulmonary neoplastic disorders with secondary pulmonary involvement, or tumors of vascular origin and neoplasms arising from supraglottic structures. Exclusion criteria included primary tumors originating in lymphatic tissue within the thoracic cavity.

### 2.1. Laboratory Findings

A routine initial workup consisted of a complete blood count, electrolytes, liver function tests, inflammatory markers (sedimentation, C-reactive protein, and lactate dehydrogenase), and urinalysis.

### 2.2. Chest Imaging and Metastasis Screening

The initial imaging tool used for each patient was a chest X-ray. Additionally, chest X-rays served for disease progression monitoring and assessment following interventional procedures.

The main indications for a chest CT scan included the persistence of radiologic lesions despite conventional antibiotic treatment or a highly suspicious mass, with or without displacement of midline structures. The procedure would be performed under analgosedation and monitored by an anesthesiologist. Based on the CT scan results, the lesions were classified into two topographic categories: intraluminal/endotracheal and peripheral/parenchymal lesions. All three tumor dimensions were expressed in millimeters, while the tissue pattern was classified as either solid or cystic/mixed based on the presence of an air–fluid level or septation within the lesion. A CT scan evaluated the vascular anatomy of the mass by comparing pre- and post-contrast series. Understanding the tumor’s regional advancement and the intrathoracic relationships revealed by the chest CT scan was crucial in determining the need for flexible bronchoscopy. The decision to proceed with transbronchial biopsy was based on CT findings. Intraluminal tumors within large airways were routinely biopsied transbronchially. Peripheral lesions associated with mediastinal shift or endobronchial extension were also considered indications for endoscopy.

The timing of follow-up chest CT scans was tailored individually, according to the treatment protocol, and typically at the end of treatment or in case of a relapse.

Abdominal ultrasound served as a tool for routine metastasis screening. When the ultrasound findings or clinical signs were suggestive of metastatic lesions elsewhere, CT or NMR would be performed.

### 2.3. Flexible Bronchoscopy (FB)

The procedure was carried out by experienced bronchologists using standard pediatric and adult flexible video bronchoscopes manufactured by Olympus Corporation (Tokyo, Japan). Each procedure was conducted under general anesthesia after obtaining informed consent from the parents. Bronchoscopic cryobiopsy and EBUS-TBNA were not accessible.

While the procedure primarily facilitated airway exploration and the collection of BALF for bacteriology and cytology, it also allowed for tissue biopsy in select cases. It provided valuable insights into the macroscopic appearance of the neoplasm, its intraluminal extension, and the degree of stenosis of the large airways. The team performed transbronchial biopsies using forceps combined with the suction catheter aspiration or needle aspiration technique, considering at least three obtained tissue samples as adequate. In addition to its diagnostic role, FB had an interventional role. Lobulated or round-shaped endobronchial lesions lacking broad bases and resembling foreign bodies were suitable for endoscopic removal and subsequent histopathological examination.

### 2.4. Treatment Modalities

Surgical resection served as the definitive treatment for patients with tumors unsuitable for endoscopic extraction or following recurrence after bronchoscopic removal. The excised tumor tissue was sent for further histopathological, immunohistochemical (IHC), and genetic analysis, making open lung biopsy the final diagnostic procedure.

In cases of unresectable lesions and metastatic disease, treatment consisted of immunotherapy, chemotherapy, and radiotherapy. The intensity, duration, and scope of these treatments depended on the type of primary lesion.

### 2.5. Histopathological Analyses

Tissue samples were processed using routine histopathological techniques, including fixation in 10% neutral-buffered formalin (Bio-Optica Milano S.p.A, Milan, Italy), paraffin embedding, and sectioning at 4 µm thickness (HistoCore MULTICUT R, Leica Biosystems, Nussloch, Germany). Hematoxylin and eosin (H&E) staining (BioGnost Ltd., Zagreb, Croatia) was performed on all samples to evaluate the morphological features (Automated Slide Stainer SS-30, Myr, Spain). For further characterization, IHC analysis was performed using a tailored panel of antibodies (Autostainer Link 48, Dako, Agilent Technologies, Santa Clara, CA, USA) selected based on the morphological features observed in H&E-stained sections to aid in diagnosis and classification. The stained slides were evaluated using bright-field optical microscopy (Olympus BX43 microscope, Olympus Corporation, Tokyo, Japan) to integrate morphological and immunophenotypic findings.

### 2.6. Genetic Analyses

In cases where the tumor presentation suggested a possible association with a broader clinical syndrome or hereditary predisposition, additional germline genetic testing was performed using peripheral blood samples. Two complementary approaches were employed: next-generation sequencing (NGS), targeting known cancer predisposition genes, and whole exome sequencing (WES), for broader variant detection across coding regions of the genome. These methods enabled high-resolution identification of pathogenic variants potentially linked to inherited tumor syndromes.

For selected tumors, following surgical resection or biopsy, additional investigation for gene variants and chromosomal rearrangements relevant to oncogenesis was performed directly on tumor tissue. Immunohistochemistry was used as part of the initial diagnostic workup, particularly to assess anaplastic lymphoma kinase (ALK) expression in cases of inflammatory myofibroblastic tumor (IMT). In ALK-negative cases, or when the tumor subtype required extended profiling, further molecular analyses were conducted. Techniques such as fluorescence in situ hybridization (FISH) and transcriptome profiling via RNA sequencing (RNA-seq) were employed to detect aberrant transcripts and fusion variants, when technically feasible and diagnostically available, with high sensitivity and specificity.

### 2.7. Statistical Analysis

The statistical analysis was performed using JASP statistical software (JASP Team (2024). JASP (Version 0.19.3) [Computer software]). Descriptive data were conveyed using frequency and traditional central tendency indicators. The Kaplan–Meier curve illustrated survival. Due to the limited sample size, conventional logistic regression was considered inappropriate, as model stability and inferential validity could not be assured. Instead, associations between clinical variables and outcomes were assessed using 2 × 2 contingency tables. In cases where cell frequencies equaled zero, the Haldane–Anscombe correction was applied by adding 0.5 to each cell. This adjustment enabled the estimation of odds ratios without relying on asymptotic assumptions, which are not suitable for sparse data. The corrected odds ratios served as indicators of effect magnitude and clinical relevance, while formal hypothesis testing was deliberately avoided due to the sample’s inadequacy for meaningful *p*-value interpretation.

## 3. Results

### 3.1. Clinical Features and Laboratory Analyses

This study examined seven patients aged two to eighteen, demonstrating a slight male predominance (4/7 or 57% male; 3/7 or 43% female). The mean age at diagnosis was 11.1 years (SD ± 5.57). The mean duration from clinical presentation to definitive diagnosis was 5.1 weeks (SD ± 4.81), with three weeks as the most common interval, occurring in 4/7 (57%) cases (Table 1).

Clinical presentations were nonspecific and heterogeneous. A dry cough was the most frequent symptom (6/7 or 86%), while other signs (fever, dyspnea, hemoptysis, and weight loss) were reported less frequently (Table 1). There were no symptoms of obstruction of the esophagus or the superior vena cava.

Routine laboratory findings were largely unremarkable. Most (5/7 or 71%) exhibited normal leukocyte counts, whereas two patients (a patient diagnosed with a carcinoid and one child with an inflammatory myofibroblastic tumor) had leukocytosis. Inflammatory markers, including C-reactive protein and lactate dehydrogenase, were elevated in 4/7 (57%) patients.

### 3.2. Chest Imaging Findings

Nonspecific, patchy parenchymal opacifications and segmental atelectasis—either unilateral or bilateral—were the most common findings, accompanied by a sagittal midline shift in four out of seven patients (57%). Hilar lymphadenopathy was present in three out of seven cases: one involving an endotracheal tumor (squamous cell carcinoma), and two involving primary parenchymal lesions (NUT carcinoma and PPB). However, chest radiographs proved insufficient for differentiating between benign and malignant lesions or for accurately assessing the extent of regional tumor spread.

In contrast, chest CT scans demonstrated high diagnostic reliability across three key parameters: identification of the primary lesion, evaluation of regional progression, and assessment of intrathoracic relationships. CT imaging revealed intraluminal airway neoplasms in four out of seven patients (57%), and primarily parenchymal neoplasms in three patients (43%) (Table 2). Intraluminal lesions arising from large airways were commonly associated with lobar or segmental atelectasis. Primary parenchymal lesions were initially unilobar, typically localized to the lower or middle lobes.

Hilar lymphadenopathy findings on CT were consistent with the chest X-ray. A solid appearance predominated in six of seven patients (86%). Notably, a patient initially diagnosed with a pulmonary abscess was ultimately found to have PPB, characterized by mixed cystic and solid components on CT imaging.

Minimal ipsilateral pleural effusion was noted in the patient with NUT carcinoma, while no effusions were present in the remaining cases. Hepatic metastases were identified in the same child.

### 3.3. Bronchologic Assessment

While each patient underwent airway exploration, 4/7 (57%) patients underwent FB with transbronchial bronchoscopy. Tumors showed two distinct endoscopic patterns:-A round-shaped and lobulated surface characterized by large airway masses with low-grade malignant potential (carcinoid and IMT);-An irregular surface with infiltrative growth and necrotic areas characterized by high-grade malignant lesions (SCC and NUT carcinoma).

No congenital structural airway abnormalities were observed. None of the patients had any major adverse events after the procedure.

### 3.4. Histopathological Findings

The histopathological analysis of the endobronchial biopsy specimen indicated the neuroendocrine origin of the tumor in a child with a carcinoid. Due to the limited size of the biopsy, tumor grading could not be determined. However, it was noted that grading would be possible on an excisional specimen. Macroscopic examination of the subsequently resected specimen revealed a polypoid tumor measuring up to 3 mm and protruding into the bronchial lumen, measuring 10 mm in diameter. Histopathological examination verified a typical carcinoid/neuroendocrine tumor, grade 1 (Figure 1). The resection margins were tumor-free.

In all three patients with IMT, the diagnosis was established from endobronchial biopsy specimens by correlating morphological features with immunohistochemical findings (Figure 2, Patient No. II).

Macroscopically, the tumor nodule in the lobectomy specimen of a child with PPB was well-defined, measuring up to 55 mm, and sharply demarcated from the surrounding lung tissue. The tumor displayed a fleshy consistency, a solid structure, and rare collapsed macrocysts. Histopathological analysis demonstrated a predominant blastemal and sarcomatoid component in solid areas, with multiple microcysts and a few larger macrocysts present (Figure 3). Correlation with additional IHC findings excluded other differential diagnoses, confirming the diagnosis of PPB type II, predominantly solid. On the mediastinal side, the tumor was covered by visceral pleura, which was focally absent, leading to positive surgical margins.

The diagnosis of NUT carcinoma was established from endobronchial biopsy specimens. Morphological analysis indicated carcinoma infiltrating the bronchial wall. The IHC profile of the tumor cells was consistent with non-small cell lung carcinoma, specifically, undifferentiated squamous cell carcinoma, with a characteristic NUT protein expression pattern in tumor cells (Figure 4).

In the child with squamous cell carcinoma of the lung, the diagnosis was made based on endobronchial biopsy samples. The degree of differentiation corresponded to a moderately differentiated tumor (Figure 5). The histogenesis of the tumor was ultimately confirmed through IHC analysis.

### 3.5. Genetic Assessment Findings

Germline genetic testing using NGS on peripheral blood samples revealed clinically significant findings in specific cases (Table 3). A patient diagnosed with pleuropulmonary blastoma was found to carry a pathogenic *DICER1* mutation, confirming a genetic predisposition identified through germline DNA sequencing. Another patient, initially diagnosed with a pituitary adenoma and later found to have a carcinoid tumor, underwent targeted genetic analysis for multiple endocrine neoplasia type 1 (MEN1). However, whole exome sequencing (WES) did not detect any pathogenic variants associated with MEN1.

Beyond its role in histopathological diagnosis, immunohistochemical analysis aided in identifying relevant somatic gene rearrangements in selected cases. Among three patients diagnosed with IMT, ALK expression was detected by immunohistochemistry in one case, supporting the presence of *ALK* gene rearrangement.

When available, molecular analyses of tumor tissue provided further insights into somatic alterations (Table 3). In the third IMT case, where ALK staining was negative, transcriptome analysis via RNA sequencing (RNA-seq) revealed a ROS1::TFG fusion, emphasizing the diagnostic value of RNA-seq in ALK-negative tumors. Due to resource constraints and the fact that analyses were conducted during different time periods, no additional molecular testing beyond immunohistochemistry was performed in one IMT patient and in a child with NUT carcinoma.

FISH analysis identified an EWSR1/ERG gene rearrangement in a resected PPB II, suggesting a role in transcriptional dysregulation.

### 3.6. Treatment and Outcomes

Various treatment modalities were applied, often in combination, to achieve complete remission. The final decision on the implemented approach depended on multiple determinants: the primary localization and type of tumor, extent of metastatic lesions, and overall health condition (Table 4).

The therapeutic role of FB was confirmed in two patients with large round-shaped airway masses: a carcinoid and one case of IMT. Removal of tumors via endoscopic biopsy forceps led to resolution of consolidated and atelectatic lung regions. Nonetheless, recurrent lesions reappeared within three months in both cases.

Radical surgical resection via lobectomy was the treatment of choice in children with pleuropulmonary blastoma and in one child with IMT, following neoadjuvant immunotherapy, with surgical resection subsequently planned. Additionally, it provided definitive remission in a child with a carcinoid tumor after recurrence following initial endoscopic removal.

Chemotherapy played a dual role. It served as the sole treatment option when other modalities were not feasible—such as NUT carcinoma, which disseminated within weeks of diagnosis. In addition, anthracycline-based chemotherapy represented the final therapeutic strategy for the child with IMT.

Neoadjuvant immunotherapy with crizotinib was used in one child with unresectable IMT exhibiting mass effect and confirmed TFG::ROS1 fusion, to induce tumor shrinkage before surgical resection.

Survival analysis was based on a limited number of events, with one patient lost to follow-up. While four out of six evaluable patients survived—representing a raw survival rate of 66.7%—Kaplan–Meier estimation, which accounts for censoring, yielded a 2-year overall survival probability of 71% (95% CI: 36–91%). This discrepancy reflects the methodological difference between simple proportions and time-to-event modeling. The resulting confidence intervals remain wide, reinforcing the need for cautious interpretation and validation in larger cohorts (Figure 6).

Using corrected odds ratios derived from contingency tables, notable associations emerged. The presence of disseminated lesions and prolonged symptom duration (>4 weeks) were each linked to an approximately 11-fold increase in the likelihood of unfavorable clinical outcomes. In addition, high-grade histological subtypes—specifically NUT carcinoma and squamous cell carcinoma—were associated with a markedly elevated risk (corrected OR ≈ 55). These effect size estimates should be interpreted as exploratory signals requiring cautious follow-up.

## 4. Discussion

While studies specifically addressing malignant pulmonary tumors in children from low-resource settings remain limited, several reports have explored these challenges within the broader context of pediatric oncology in low- and middle-income countries (LMICs) [13,14,15]. The limited availability of advanced diagnostic tools continues to pose a significant barrier to timely and accurate diagnosis. Although the World Health Organization (WHO) has recommended a core list of 100 essential diagnostic items for pediatric oncology in LMICs, access to even these fundamental resources remains inconsistent across many regions [13,14,15]. In particular, access to molecular profiling technologies—especially those detecting gene rearrangements essential for tumor proliferation and survival—is frequently restricted. This limitation compromises accurate tumor classification and hinders the implementation of personalized treatment strategies [13,14,15,16]. Diagnostic delays, often stemming from prolonged prediagnostic intervals, further contribute to poorer clinical outcomes [13,14,16]. In our cohort, prolonged symptom duration was associated with an approximately 11-fold increase in the likelihood of an unfavorable clinical outcome (corrected OR ≈ 11.0), reinforcing the detrimental impact of delayed diagnosis even at the individual case level. Moreover, insufficient awareness among healthcare providers and the general public can result in both underdiagnosis and misdiagnosis [13]. Although population-wide screening for pediatric cancers remains controversial, risk-based surveillance strategies may facilitate earlier detection in genetically predisposed individuals. In this context, the identification of a pathogenic *DICER1* mutation in our study highlights a promising shift toward incorporating molecular surveillance in pediatric patients at increased oncologic risk.

The heterogeneous clinical presentation of primary malignant lung tumors, along with their radiographic characteristics, often precludes a definitive distinction from non-neoplastic changes [17]. This uncertainty necessitates the need for histopathological analysis of biopsy samples, which is crucial for establishing a definitive diagnosis that guides treatment and predicts the prognosis of the disease [18].

While chest X-ray remains a widely accessible and commonly used initial imaging tool in pediatric respiratory evaluation, its diagnostic value for lung tumors is limited [19]. Although it may detect mass lesions and associated complications such as atelectasis, pleural effusion, or pneumothorax, it often fails to reliably distinguish between neoplastic and non-neoplastic processes. Moreover, chest X-ray does not provide insight into tumor vascularization or precise spatial relationships within the thoracic cavity—unless a clear mass effect is present [19,20]. Importantly, it offers no information regarding the nature or histological subtype of the lesion. Many of these shortcomings are addressed by chest CT.

Although chest CT scans cannot reliably indicate the type of tumor, this procedure is paramount for assessing the size of the lesion, its relationship with surrounding structures, and the origin of its blood supply [19]. Particular focus is placed on identifying characteristics that distinguish malignant and benign lung tumors. While bilateral distribution strongly suggests malignancy, factors such as tumor size, morphological appearance, and hilar or mediastinal lymphadenopathy are not consistently reliable indicators [19]. Metastatic involvement of both regional and distant lymph nodes may occur even in the absence of radiologically apparent lymphadenopathy [21]. Furthermore, malignant lung tumors may initially be misinterpreted as non-neoplastic changes on CT imaging, potentially delaying diagnosis and increasing the risk of poor clinical outcomes [22]. The reliability of preoperative diagnosis in malignant lesions depends on the imaging modality, the radiologist’s level of expertise, and the underlying pulmonary pathology. Reported overall diagnostic accuracy rates below 80%—particularly in distinguishing PPB from congenital pulmonary airway malformation—present significant challenges and underscore the need for a comprehensive, multi-modal diagnostic approach [23]. To reduce the risk of delayed diagnosis in cases of malignancy and to avoid unnecessary invasive procedures in benign or non-neoplastic conditions, integrated algorithms have been introduced [18,24]. These approaches combine radiologic features—sagittal displacement of mediastinal structures and lesion vascularity—with bronchoscopic assessment and serum-based genetic testing [18,24]. In type I PPB, this approach has resulted in nearly 100% positive and negative predictive values for distinguishing it from congenital pulmonary airway malformation type IV [25]. Unfortunately, diagnostic algorithms for solid tumor formations are not as widely developed as those for cystic lesions. Thus, a tissue sample is subjected to histological investigation to establish a definitive diagnosis. Presuming that surgical resection is the most common form of definitive treatment, postoperative tissue analysis remains the gold standard for final diagnosis [23]. Although reliable, this method of determining the nature of observed changes has its drawbacks. Primarily, it involves an invasive surgical approach and, by its nature, excludes definitive knowledge of the neoplasm type in the preoperative period.

Flexible bronchoscopy has led to a significant advancement in the diagnostic domain [26,27]. The minimally invasive nature of flexible bronchoscopy, coupled with technical advancements that have reduced procedure-related adverse effects, allows for the visualization of endoluminal propagation and the macroscopic appearance of the tumor mass [26]. In our series, there appeared to be a correlation between the macroscopic appearance of the tumor and the degree of malignancy. Specifically, a lobulated structure with a smooth surface was associated with low-grade malignances, whereas an irregular surface with infiltrative growth suggested high-grade tumors. However, the current literature does not support the reliability of macroscopic appearance as a standalone method for assessing malignant potential [28]. Nevertheless, visual assessment of the endoluminal mass contributes to evaluating the feasibility of endoscopic tumor extraction, potentially avoiding the need for more aggressive surgical approaches. Notably, endoscopic extraction may be considered in selected cases where the tumor is unifocal, non-infiltrative, and not associated with lymph node involvement [8,29]. As recent bronchoscopic advancements have made this approach safer, the risk of tumor regrowth remains a concern and warrants definitive treatment with surgical excision or chemotherapy [8,29].

While limited access to advanced diagnostic resources is a well-recognized issue in LMICs, growing attention is being directed toward the widening gap in therapeutic capacity—particularly in the context of precision medicine [30,31,32]. The availability and affordability of essential chemotherapeutic agents, and more notably, targeted therapies, remain inconsistent and often prohibitively limited. Although actionable molecular alterations are increasingly identified through institutional and collaborative initiatives, access to matched targeted treatments remains a critical obstacle [13,33]. Contributing factors include the off-label status of many targeted agents in pediatric populations, lack of regulatory approval or national formulary inclusion, and insufficient infrastructure to ensure safe drug administration and monitoring. Nonetheless, international efforts such as virtual molecular tumor boards and multicenter collaborations have enabled select LMICs to begin integrating molecular insights into therapeutic planning [34,35]. Over time, the focus in these settings has begun shifting from the absence of universal protocols to the development of context-sensitive, biomarker-guided treatment strategies [30,31,33]. These developments highlight the indispensable role of global partnerships, coordinated clinical trials, and standardized therapeutic frameworks in reducing pediatric oncology disparities worldwide.

In light of the increasing relevance of molecular diagnostics in pediatric lung tumors, it is equally important to consider the potential syndromic and hereditary contexts in selected cases. While most tumors arise sporadically, certain histological subtypes may serve as early indicators of broader genetic syndromes or familial cancer predisposition [36,37].

Typical carcinoid tumors, for example, though usually isolated, may occur within the spectrum of multiple endocrine neoplasia type 1 (MEN1). In such cases, targeted genetic testing is warranted to evaluate the risk of associated endocrine neoplasms and to guide longitudinal follow-up [36]. Similarly, pleuropulmonary blastoma (PPB) may be the first clinical manifestation in individuals with germline *DICER1* mutations, which predispose to a range of neoplastic conditions. When a child is identified as the proband, cascade testing of family members is recommended, forming the foundation for early cancer screening and risk stratification [37]. In both scenarios, genetic counseling is essential to ensure appropriate interpretation of test results, facilitate informed decision-making, and support families in navigating surveillance and preventive strategies.

As the detection of somatic rearrangements such as EWSR1/ERG and TFG::ROS1 illustrates the utility of molecular profiling in uncovering oncogenic drivers, numerous molecular findings are still of unknown significance. Concomitant detection of the EWSR1/ERG gene rearrangement in a patient with type II PPB has not been described so far in this type of tumor. Typically involved in Ewing sarcoma genesis, this gene fusion is the subject of thorough investigations, with the potential to become a target for innovative therapy [38].

The discovery of ROS1 fusion rearrangements—most notably the TFG::ROS1 oncoprotein—has transformed our understanding of inflammatory myofibroblastic tumor (IMT) biology [39]. By constitutively activating the ROS1 kinase domain, these chimeric proteins drive tumor growth and are linked to a more aggressive clinical course, giving them both diagnostic and prognostic weight [40]. Comprehensive genomic surveys demonstrate that roughly 85% of IMTs harbor kinase fusions, with *ALK* alterations most common; in *ALK*-negative cases, systematic screening for ROS1 rearrangements is therefore essential [40]. Clinically, IMTs positive for TFG::ROS1 show poor response to standard anti-inflammatory and cytotoxic chemotherapy, but may achieve significant and durable partial remissions with crizotinib. In pediatric reports, children whose tumors carried TFG::ROS1 experienced marked shrinkage and symptom relief within months of starting crizotinib, underscoring the need for early molecular testing to guide targeted therapy [39,40].

From a therapeutic standpoint, key questions persist regarding the optimal timing for introducing crizotinib in ROS1-positive IMT. While early reports suggest benefit in the neoadjuvant setting, it remains unclear how long targeted therapy should be administered and whether it has a role beyond initial tumor reduction [39]. The potential emergence of resistance to first-generation ROS1 inhibitors may limit long-term efficacy and complicate future treatment strategies [41].

Moreover, while high-income centers increasingly employ broad fusion panels to capture ALK, ROS1, and other kinase rearrangements, many resource-constrained institutions still test only for ALK, delaying or missing ROS1-driven cases altogether [13,41]. This gap not only hinders patient access to novel agents—currently off-label in most pediatric IMTs—but also limits the feasibility of biomarker-driven clinical trials aimed at improving outcomes in this rare disease.

This study has several limitations. Its small study population, retrospective design, single-center bias, and relative heterogeneity in terms of tumor etiology and histopathological features necessitate cautious interpretation of the findings. Moreover, the brevity of clinical follow-up hindered a meaningful assessment of long-term survival, as the available data lacked sufficient temporal depth to support reliable estimates beyond early outcome horizons. Additionally, an important limitation was the absence of comprehensive molecular and genetic profiling of tumor tissue in all patients. The investigation of mutations and gene rearrangements was not uniformly performed, primarily due to resource constraints and the fact that diagnoses were established across different time frames. As a result, certain potentially actionable alterations may have remained undetected. Finally, it should be emphasized that the conclusions presented herein are based on a case series and do not constitute formal, internationally validated treatment recommendations.

## 5. Conclusions

Primary malignant lung tumors in children remain rare and diagnostically challenging entities that often mimic nonmalignant pulmonary conditions. Accurate diagnosis relies on the combined use of radiologic, bronchoscopic, histopathological, and molecular-genetic tools. In this study, the identification of an EWSR1::ERG gene rearrangement in a case of pleuropulmonary blastoma—a finding not previously reported in this tumor type—underscores the evolving role of molecular profiling in refining diagnosis and understanding tumor biology. Additionally, the detection of a TFG::ROS1 fusion in an ALK-negative inflammatory myofibroblastic tumor and the neoadjuvant use of crizotinib in that patient demonstrate the expanding relevance of targeted therapies in pediatric pulmonary malignancies. While conventional treatment approaches continue to offer curative outcomes in many cases, the integration of precision medicine strategies holds promise for improving outcomes in selected patients with high-risk or treatment-refractory disease.

## Figures and Tables

**Figure 1 biomedicines-13-01824-f001:**
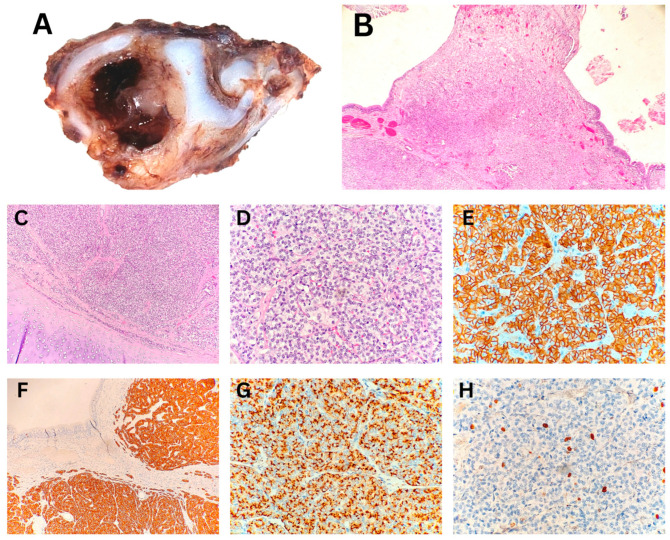
Typical carcinoid/neuroendocrine tumor, grade 1—excisional biopsy. (**A**) Macroscopic cross-section of the bronchus showing a polypoid tumor within the lumen. (**B**) Ulcerated polypoid tumor infiltrating the bronchial wall (40× magnification). (**C**) Tumor infiltration extending to the boundary with the cartilage (100× magnification). (**D**) Organoid nests and trabeculae of tumor cells with “salt-and-pepper” chromatin and low mitotic activity (400× magnification). (**E**,**F**,**G**) Immunohistochemical positivity for CD56, Synaptophysin, and Chromogranin, respectively (400×, 100× and 400× magnification). (**H**) Proliferative marker Ki-67 is positive in fewer than 4–5% of tumor cells (400× magnification).

**Figure 2 biomedicines-13-01824-f002:**
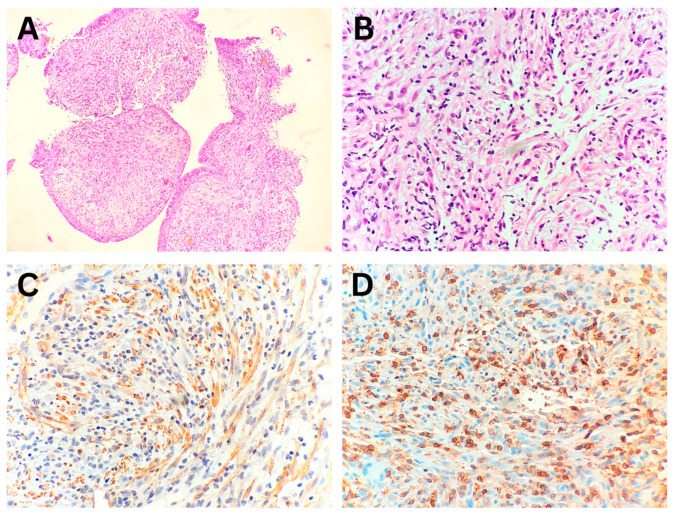
Inflammatory myofibroblastic tumor of the lung—endobronchial biopsy. (**A**) Bronchial wall with surface squamous metaplasia infiltrated by tumor tissue (40× magnification). (**B**) Uniform spindle-shaped tumor cells with myofibroblastic morphology intermixed with inflammatory cells (400× magnification). (**C**) Immunohistochemical positivity of tumor cells for smooth muscle actin (400× magnification). (**D**) Immunohistochemical positivity of inflammatory cells for leukocyte common antigen (400× magnification).

**Figure 3 biomedicines-13-01824-f003:**
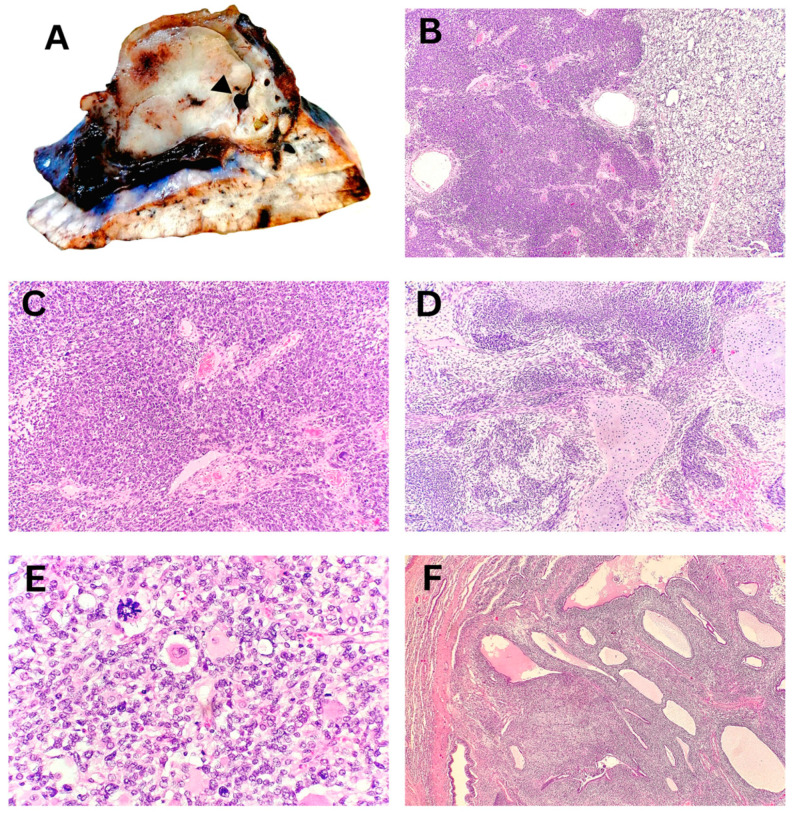
Pleuropulmonary blastoma, type II, predominantly solid—excisional biopsy. (**A**) Macroscopic appearance of a solid tumor with a collapsed macrocytic area (arrowhead). (**B**) Solid tumor component composed of blastematous and sarcomatoid elements (40× magnification). (**C**) Primitive, undifferentiated blastematous cells with numerous mitotic figures (100× magnification). (**D**) Sarcomatoid fibromyxoid component with foci of chondroid differentiation (100× magnification). (**E**) Zones of anaplasia with atypical mitotic figures and rhabdomyoblastic differentiation (400× magnification). (**F**) Multiple microcysts surrounded by condensed blastematous cells (40× magnification).

**Figure 4 biomedicines-13-01824-f004:**
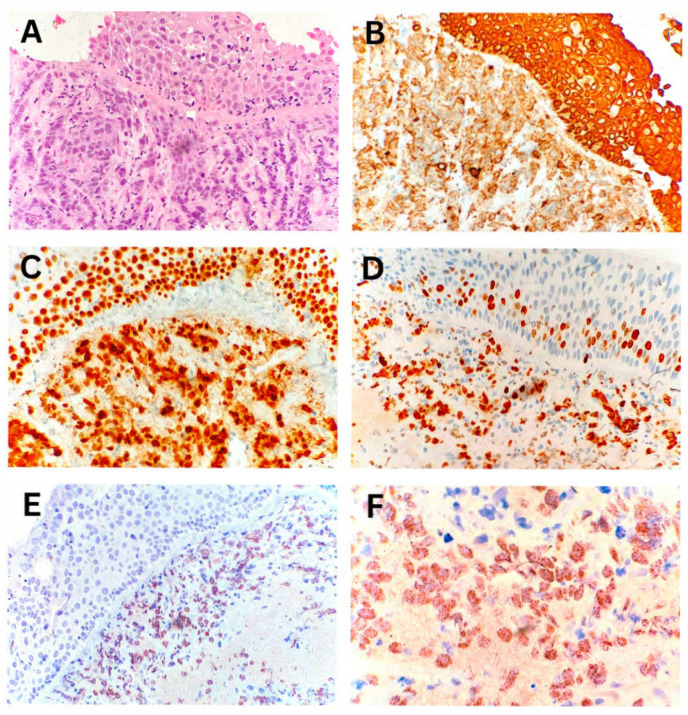
NUT carcinoma of the lung—endobronchial biopsy. (**A**) Bronchial wall with surface squamous metaplasia infiltrated by tumor tissue (400× magnification). (**B**) Immunohistochemical staining for cytokeratin AE1/AE3 showing cytoplasmic positivity in metaplastic squamous epithelium and subepithelial tumor cells (400× magnification). (**C**) Immunohistochemical staining for p63 showing nuclear positivity in the same regions (400× magnification). (**D**) Proliferative marker Ki-67 is positive in approximately 60% of tumor cells (400× magnification). (**E**,**F**) Characteristic nuclear dot-like positivity of tumor cells for NUT protein (400× and 1000× magnification).

**Figure 5 biomedicines-13-01824-f005:**
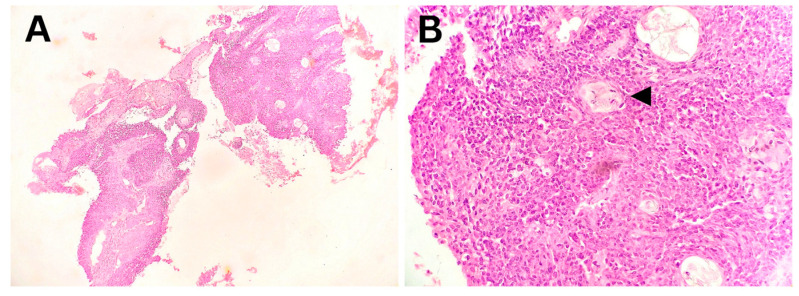
Squamous cell carcinoma of the lung, moderately differentiated—endobronchial biopsy. (**A**) Tissue fragments with nests of infiltrative tumor growth (40× magnification). (**B**) Smaller epithelial tumor cells with a focus of squamous differentiation (arrowhead, 400× magnification).

**Figure 6 biomedicines-13-01824-f006:**
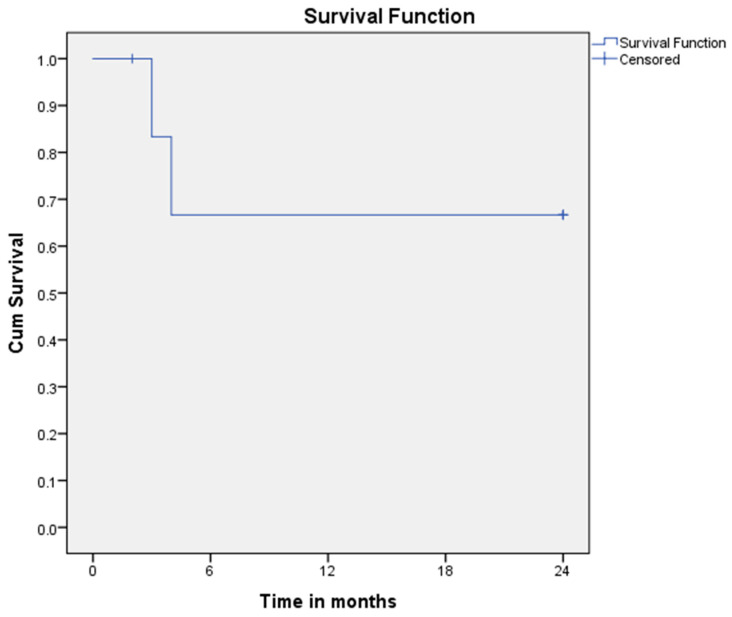
Kaplan–Meier survival curve over a two-year period.

**Table 1 biomedicines-13-01824-t001:** Demographics and clinical features.

Patient	Sex	Age at Dg	Time to Dg	Cough	Fever	Dyspnea	Hemoptysis	Weight Loss	Type of a Tumor
I	M	11 y	3 weeks	+	+	−	−	−	Carcinoid
II	F	9 y	3 weeks	+	−	+	−	−	IMT
III	F	2.5 y	3 weeks	+	+	−	−	−	PPB II
IV	M	16 y	4 weeks	+	−	−	+	+	NC
V	M	18 y	4 months	+	−	+	+	+	SCC
VI	M	6.5 y	4 weeks	+	+	+	−	+	IMT
VII	M	15 y	3 weeks	−	−	+	−	−	IMT

M—male; F—female; Y—years; IMT—inflammatory myofibroblastic tumor; PPB II—pleuropulmonary blastoma type II; NC—NUT carcinoma; SCC—squamous cell carcinoma.

**Table 2 biomedicines-13-01824-t002:** Radiological features of primary lung tumors.

	Chest X−Rays			Chest CT Scan			
	Lobar/Segmental Atelectasis	Hilar Adenopathy	Mediastinal Shift	Origin	Localization	Appearance	Hilar Adenopathy
I	+	−	−	Airways	Intermediate bronchus	Solid	−
II	+	−	+	Airways	Main tracheal carina	Solid	−
III	−	+	−	Parenchyma	Left lower lobe	Mixed Cystic/solid	+
IV	+	+	+	Parenchyma	Middle lobe	Solid	+
V	−	+	+	Airways	Right upper lobe bronchus	Solid	+
VI	+	−	+	Parenchyma	Right lower lobe	Solid	+
VII	+	−	−	Airways	Left main bronchus	Solid	−

**Table 3 biomedicines-13-01824-t003:** Summary of genetic findings, diagnostic methods, and targeted therapy in patients with primary pulmonary tumors.

PH Diagnosis	Genetic Assessment		Diagnostic Method	Target Therapy
	Peripheral Blood	Tumor Tissue		
Carcinoid	MEN1 −	-	WES *	-
IMT	-	ALK −	IHC staining **	-
PPB II	DICER1 +	EWSR1/ERG +	NGS * RNA-seq **	-
NC	-	-	-	-
SCC	-	-	-	-
IMT	-	ALK − TFG::ROS1 +	IHC staining ** RNA-seq **	Crizotinib
IMT	-	ALK +	IHC staining **	-

IMT—inflammatory myofibroblastic tumor; PPB II—pleuropulmonary blastoma type II; NC—NUT carcinoma; SCC—squamous cell carcinoma; MEN1—multiple endocrine neoplasia type 1 gene; ALK—Anaplastic Lymphoma Kinase; WES—whole exome sequencing; IHC—immunohistochemical staining; NGS—next-generation sequencing; RNA-seq—RNA sequencing. * Diagnostic methods used for genetic testing of peripheral blood samples. ** Diagnostic methods used for genetic testing of tissue specimens.

**Table 4 biomedicines-13-01824-t004:** Treatment modalities and outcomes.

	Tumor	Initial Treatment	Recurrent Lesion	Definitive Treatment	Outcome
I	Carcinoid	Endoscopic extraction	Yes	Surgical resection	Alive
II	IMT	Endoscopic extraction	Yes	Chemotherapy	Alive
III	PPB II	Surgical resection	No	-	Alive
IV	NC	Chemotherapy	-	-	Death
V	SCC	Refused treatment	-	-	Death
VI	IMT	Immunotherapy	No	Surgical resection *	Alive
VII	IMT	Surgical resection	No	-	Lost to follow-up

IMT—inflammatory myofibroblastic tumor; PPB II—pleuropulmonary blastoma type II; NC—NUT carcinoma; SCC—squamous cell carcinoma. * Surgical resection was planned but had not been performed at the time of reporting.

## Data Availability

The original contributions presented in this study are included in the article. Further inquiries can be directed to the corresponding author.

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
