# Peer review of "Clinical Characteristics, Diagnosis, and Management of Primary Malignant Lung Tumors in Children: A Single-Center Analysis"

_biomedicines, 2025, doi:10.3390/biomedicines13081824_

Round 1
Reviewer 1 Report
Comments and Suggestions for Authors
In the manuscript entitled “Clinical Characteristics, Diagnosis, and Management of Primary Malignant Lung Tumors in Children: A Single-Center Analysis,” the authors offered a valuable retrospective analysis of primary malignant lung tumors in children—a rare and underrepresented topic. The authors provide detailed diagnostic pathways and treatment strategies, which can guide future clinical management, especially in resource-limited settings.
The integration of radiological, bronchoscopic, histopathological, and genetic testing is a major strength. Highlighting the utility of advanced techniques like EBUS-TBNA and RNA-seq for fusion gene detection adds scientific depth and clinical relevance. Their work made valuable contributions to a rare pediatric entity. However, there are some concerns should be considered, as listed in the following points:
- Clear Presentation of Case Series with Useful Clinical Data
The clinical data are well-organized, with meaningful summaries of patient characteristics, presentation, diagnosis, and treatment. However, including a more detailed summary table (age, tumor type, treatment, outcome) would improve readability. - Limited Sample Size and Generalizability
While the case series offers detailed insight, the small cohort (n=7) limits statistical conclusions and generalizability. The authors may consider acknowledging this more explicitly in the discussion and contextualizing findings with available international literature. - Genomic Findings Need Expanded Interpretation
The identification of a DICER1 mutation and a TFG::ROS1 fusion are important findings, but their implications for targeted therapy could be discussed more thoroughly. Are these routinely screened in similar tumors? Could this support future biomarker-driven clinical trials? - Ethical Approval and Consent Properly Addressed
The manuscript clearly states that ethics approval and parental consent were obtained, which ensures compliance with publication and research ethics.
Author Response
Comments and Suggestions for Authors
In the manuscript entitled “Clinical Characteristics, Diagnosis, and Management of Primary Malignant Lung Tumors in Children: A Single-Center Analysis,” the authors offered a valuable retrospective analysis of primary malignant lung tumors in children—a rare and underrepresented topic. The authors provide detailed diagnostic pathways and treatment strategies, which can guide future clinical management, especially in resource-limited settings.
The integration of radiological, bronchoscopic, histopathological, and genetic testing is a major strength. Highlighting the utility of advanced techniques like EBUS-TBNA and RNA-seq for fusion gene detection adds scientific depth and clinical relevance. Their work made valuable contributions to a rare pediatric entity. However, there are some concerns should be considered, as listed in the following points:
- Clear Presentation of Case Series with Useful Clinical Data
The clinical data are well-organized, with meaningful summaries of patient characteristics, presentation, diagnosis, and treatment. However, including a more detailed summary table (age, tumor type, treatment, outcome) would improve readability.
Response:
Thank you for your constructive and helpful comment. In addition to Table 1, which already summarizes tumor types, demographic data, and clinical presentation, a new table (Table 3) has now been added to further enhance the clarity of the results. This table presents details on the genetic mutations and rearrangements investigated in individual patients, the diagnostic modalities used to detect them, and targeted therapies aimed at specific signaling pathways that were applicable within our institutional setting. We believe this addition has improved the readability and completeness of the dataset. Consequently, the previously numbered Table 3 is now designated as Table 4. The new table is located in lines 332–341 of the revised manuscript.
- Limited Sample Size and Generalizability
While the case series offers detailed insight, the small cohort (n=7) limits statistical conclusions and generalizability. The authors may consider acknowledging this more explicitly in the discussion and contextualizing findings with available international literature.
Response:
We fully recognize that the small cohort size (n = 7) inherently limits statistical power and generalizability of the findings. In response, a dedicated paragraph addressing these limitations has been added to the manuscript (lines 539–550), emphasizing the constrained interpretability of survival data and prognostic factors. Furthermore, the Treatment and Outcomes section (lines 383–396) has been revised to explicitly highlight the need for caution when drawing conclusions from this preliminary dataset. These additions aim to contextualize our findings within the boundaries imposed by sample size and underscore the importance of further investigation in larger cohorts.
- Genomic Findings Need Expanded Interpretation
The identification of a DICER1 mutation and a TFG::ROS1 fusion are important findings, but their implications for targeted therapy could be discussed more thoroughly. Are these routinely screened in similar tumors? Could this support future biomarker-driven clinical trials?
Response:
Thank you for highlighting this important aspect. The manuscript has been substantially revised to incorporate a more in-depth discussion of the molecular findings and their relevance for targeted therapy.
-A dedicated subsection (2.6 Genetic Analyses, lines 194–211) now outlines the methodology used for detecting mutations and rearrangements, including DICER1 and TFG::ROS1, and the corresponding diagnostic modalities.
-In the Results section, a new subsection titled 3.5 Genetic Assessment Findings (lines 318–350) has been introduced, integrating previous content and presenting a detailed overview of the identified genomic alterations and their therapeutic implications.
-The Discussion section has been thoroughly revised and expanded (lines 494–538) to address the clinical relevance of these findings and their potential to guide biomarker-informed treatment decisions in pediatric pulmonary malignancies.
We believe these additions provide a more comprehensive and translational perspective on the molecular profile of the cohort and its potential utility in future biomarker-driven clinical trials.
- Ethical Approval and Consent Properly Addressed
The manuscript clearly states that ethics approval and parental consent were obtained, which ensures compliance with publication and research ethics.
Reviewer 2 Report
Comments and Suggestions for Authors
- Better to describe types of malignant lung tumors in the introduction section.
- In my view, abstract should be concise. Consider emphasizing the novel findings (e.g. ROS1, EWSR1 fusions, crizotinib use) to increase the impact.
- Provide a summary table detailing the targeted gene alterations identified in this study including the genes implicated, detection methods, and therapeutic relevance.
- Some sentences and phrases are repeated.
- Sample size is very small. Include the limitations of the study regarding small sample size, single center-bias, limited molecular profiling due to resource constraints.
- Clarify the diagnostic methods used to detect genetic alterations. What were the diagnostic methods used for RNA sequencing?
- What was the technique used for confirmation of NUTM1 gene arrangement?
- The histological confirmation of typical carcinoid is very satisfied. However, a discussion on the lack of mutations implicated in typical carcinoid in pediatric patients would enhance the completeness of the section.
- Briefly acknowledge the context on hereditary predisposition and genetic counseling relevance in discussion section.
- Highlight the clinical decision-making following genetic confirmation for IMT and PPB.
- The authors are advised to briefly elaborate on the limitations of chest X-Ray and CT imaging in pediatric lung tumor diagnosis.
- Consider the place of comma, full stop and break complex sentences into short sentences.
- Conclusion: The authors are encouraged to highlight the novel genetic findings and the significant targeted therapies to enhance the clinical relevance of this study.
Whole manuscript should be checked for syntax, comma, full stop.
Author Response
Reviewer 2
Comments and Suggestions for Authors
- Better to describe types of malignant lung tumors in the introduction section.
Response:
Thank you for your valuable suggestion. In response, the Introduction section has been expanded to include a concise overview of the most common malignant lung tumors observed in pediatric patients. This new content can be found in lines 58–72 of the revised manuscript and provides contextual framing for the clinical cases that follow. We appreciate your insight, which helped improve the clarity and completeness of the background information.
- In my view, abstract should be concise. Consider emphasizing the novel findings (e.g. ROS1, EWSR1 fusions, crizotinib use) to increase the impact.
Response:
Thank you for this valuable suggestion. In response, the abstract has been restructured to better reflect the scope and intent of the manuscript (lines 28–51). The revised version provides a more concise overview of the study while emphasizing its novel findings. Specifically, the identification of a ROS1::TFG fusion in one patient and the subsequent administration of targeted therapy with crizotinib are now clearly highlighted (lines 44–46), enhancing the clinical relevance and impact of the report.
- Provide a summary table detailing the targeted gene alterations identified in this study including the genes implicated, detection methods, and therapeutic relevance.
Response:
In addition to Table 1, which already summarizes tumor types, demographic data, and clinical presentation, a new table (Table 3) has now been added to further enhance the clarity of the results. This table presents details on the genetic mutations and rearrangements investigated in individual patients, the diagnostic modalities used to detect them, and targeted therapies aimed at specific signaling pathways that were applicable within our institutional setting. We believe this addition has improved the readability and completeness of the dataset. Consequently, the previously numbered Table 3 is now designated as Table 4. The new table is located in lines 332–341 of the revised manuscript.
- Some sentences and phrases are repeated.
Response:
In response to your thoughtful observation, parts of the Materials and Methods and Results sections have been restructured to reduce redundancy and improve clarity. Several repetitive sentences were removed, and minor stylistic and linguistic edits were applied throughout the section. These revisions have contributed to a more concise and focused presentation of diagnostic and therapeutic findings. All modifications are visible in the tracked changes, spanning multiple subsections within Materials and Methods and Results.
- Sample size is very small. Include the limitations of the study regarding small sample size, single center-bias, limited molecular profiling due to resource constraints.
Response:
We fully recognize that the small cohort size (n = 7) inherently limits statistical power and generalizability of the findings. In response, a dedicated paragraph addressing these limitations has been added to the manuscript (lines 539–550), emphasizing the constrained interpretability of survival data and prognostic factors. Furthermore, the Treatment and Outcomes section (lines 383–396) has been revised to explicitly highlight the need for caution when drawing conclusions from this preliminary dataset. These additions aim to contextualize our findings within the boundaries imposed by sample size and underscore the importance of further investigation in larger cohorts.
- Clarify the diagnostic methods used to detect genetic alterations. What were the diagnostic methods used for RNA sequencing?
Response:
The manuscript has been substantially revised to include a more detailed account of the diagnostic methods used to detect genetic alterations and their clinical relevance.
- A dedicated subsection (2.6 Genetic Analyses, lines 194–211) now outlines the molecular techniques employed, including RNA sequencing methods used to identify mutations and rearrangements such as DICER1 and ROS1::TFG.
- In the Results section, a new subsection titled 3.5 Genetic Assessment Findings (lines 318–350) has been introduced, integrating molecular results with therapeutic implications.
- These elements are now complemented by a new summary table (Table 3), which consolidates genetic findings and associated diagnostic approaches.
- What was the technique used for confirmation of NUTM1 gene arrangement?
Response:
The diagnosis of NUT carcinoma in our patient was established based on endobronchial biopsy specimens. Morphologic assessment revealed infiltration of the bronchial wall by malignant cells, and immunohistochemical (IHC) analysis demonstrated a characteristic expression pattern of NUT protein within tumor cells. However, techniques for direct confirmation of NUTM1 gene rearrangement—such as fluorescence in situ hybridization (FISH) or RNA sequencing—were not employed in this case.
Additionally, as now stated in the Limitations section (lines 539-550), access to advanced molecular diagnostic tools was constrained, partly due to the retrospective nature of case inclusion and temporal variability in diagnostic resources. This limitation affected our ability to comprehensively confirm genetic findings across all patients.
- The histological confirmation of typical carcinoid is very satisfied. However, a discussion on the lack of mutations implicated in typical carcinoid in pediatric patients would enhance the completeness of the section.
AND - Briefly acknowledge the context on hereditary predisposition and genetic counseling relevance in discussion section.
Response:
Thank you for highlighting these complementary aspects. A single, integrated paragraph addressing both the mutational context of typical carcinoid tumors and the relevance of hereditary predisposition has been added to the Discussion section (lines 494–508). Given the conceptual proximity of these themes, we considered it most appropriate to treat them jointly rather than in isolation. We trust this focused revision enhances the clinical and genetic depth of the discussion while maintaining clarity.
- Highlight the clinical decision-making following genetic confirmation for IMT and PPB.
Response:
While the topic of hereditary predisposition and typical carcinoid mutations was addressed in a unified paragraph within the Discussion section (lines 494–508), a separate and more detailed segment now focuses specifically on IMT (lines 515–538).
This newly added section discusses the therapeutic implications of ROS1 fusion-positive IMTs—including diagnostic pathways, the rationale for early molecular screening, and the role of targeted therapy with crizotinib. It also highlights challenges in clinical implementation, such as timing of therapy, potential resistance mechanisms, and disparities in access to comprehensive fusion testing. Together, these additions aim to contextualize the molecular findings within a decision-making framework and support translational relevance in pediatric oncology.
- The authors are advised to briefly elaborate on the limitations of chest X-Ray and CT imaging in pediatric lung tumor diagnosis.
Response:
The Discussion section has been expanded to include a dedicated segment addressing the limitations of chest X-ray imaging in the context of pediatric lung tumor diagnosis (lines 425–432). This addition complements the existing paragraph on CT limitations (lines 435–453), and together they provide a more balanced and comprehensive overview of the diagnostic challenges associated with radiologic modalities in this patient population.
- Consider the place of comma, full stop and break complex sentences into short sentences.
Response:
We have thoroughly reviewed the entire manuscript with particular attention to punctuation (commas and full stops) and sentence structure. Several overly long or complex sentences were revised and divided into shorter, clearer units to improve readability and flow. We also made fine adjustments to comma placement to ensure proper syntax and adherence to academic writing conventions. These corrections were applied consistently throughout the text, and we believe the overall clarity of the manuscript has been significantly improved as a result.
- Conclusion: The authors are encouraged to highlight the novel genetic findings and the significant targeted therapies to enhance the clinical relevance of this study.
Response:
We agree with the reviewer’s suggestion. The Conclusion section has been revised to highlight the novel EWSR1::ERG gene rearrangement identified in pleuropulmonary blastoma, as well as the relevant use of targeted therapy (crizotinib) in an ALK-negative IMT case with a TFG::ROS1 fusion (lines 551-564). These additions enhance the clinical relevance of the study, as recommended.
Reviewer 3 Report
Comments and Suggestions for Authors
I present my review in subsections, which should be addressed in the response to the reviewer and, if possible, improve and enhance the text of the article.
- The paper claims to offer a diagnostic and treatment strategy for under-resourced healthcare systems but relies on a single-center, retrospective review of only seven patients. Could the authors provide stronger justification for generalizing these conclusions, or more clearly limit their scope?
- The results mention a two-year survival rate of 66.7% (based on 4 survivors out of 6 with known outcomes), but there is no discussion of confidence intervals or analysis of prognostic factors. Could the authors clarify these limitations or provide more detail about the reliability of the Kaplan-Meier analysis in such a small cohort?
- Parts of the Results (e.g., bronchoscopic and histopathological descriptions) are rich in detail but contain some repetition. Would the authors consider restructuring or summarizing key diagnostic and therapeutic findings more concisely—perhaps with an additional table?
- The paper emphasizes the value of genetic testing and targeted therapies (e.g., crizotinib in IMT with TFG::ROS1 fusion), but it lacks discussion of their availability and cost in resource-limited settings—the very context the paper claims to address. Could the authors expand on these practical considerations?
- The summary and conclusions seem to offer prescriptive strategies, but the data are essentially from an uncontrolled series of cases. Would the authors consider clarifying that these are observations from a case series rather than formal, validated recommendations?
Author Response
Comments and Suggestions for Authors
I present my review in subsections, which should be addressed in the response to the reviewer and, if possible, improve and enhance the text of the article.
- The paper claims to offer a diagnostic and treatment strategy for under-resourced healthcare systems but relies on a single-center, retrospective review of only seven patients. Could the authors provide stronger justification for generalizing these conclusions, or more clearly limit their scope?
Response:
We trully appreciate this important observation! In response, we revised the phrasing in the Introduction to more accurately reflect the nature and scope of our study. Instead of stating that the study “proposes a diagnostic and treatment strategy,” we now state that it “illustrates a pragmatic diagnostic and therapeutic approach” (lines 101-103). This change avoids implying generalizable or prescriptive recommendations and instead emphasizes that the findings are based on a small, retrospective case series and are intended to inform, not dictate, clinical decision-making—particularly in resource-constrained settings where formal guidelines may be lacking.
- The results mention a two-year survival rate of 66.7% (based on 4 survivors out of 6 with known outcomes), but there is no discussion of confidence intervals or analysis of prognostic factors. Could the authors clarify these limitations or provide more detail about the reliability of the Kaplan-Meier analysis in such a small cohort?
Response:
In accordance with your valuable suggestion, the manuscript has been revised and expanded to include additional data and methodological clarifications:
- First, in Section 2.7 (Statistical Analysis, lines 218–226), a detailed description of the analytical approach used to evaluate prognostic factors affecting survival has been added.
Due to the limited sample size, conventional logistic regression was considered inappropriate, as model stability and inferential validity could not be assured. Instead, associations between clinical variables and outcomes were assessed using 2×2 contingency tables. In cases where cell frequencies equaled zero, the Haldane–Anscombe correction was applied by adding 0.5 to each cell. This adjustment enabled the estimation of odds ratios without relying on asymptotic assumptions, which are not suitable for sparse data. The corrected odds ratios served as indicators of effect magnitude and clinical relevance, while formal hypothesis testing was deliberately avoided due to the sample’s inadequacy for meaningful p-value interpretation.
- Furthermore, in theResults section (lines 380–396), the Kaplan–Meier survival curve is now accompanied by corresponding confidence intervals, along with a brief commentary on the limitations of this approach given the small cohort size.
Survival analysis was based on a limited number of events, with one patient lost to follow-up. While 4 out of 6 evaluable patients survived—representing a raw survival rate of 66.7%—Kaplan–Meier estimation, which accounts for censoring, yielded a 2-year overall survival probability of 71% (95% CI: 36–91%). This discrepancy reflects the methodological difference between simple proportions and time-to-event modeling. The resulting confidence intervals remain wide, reinforcing the need for cautious interpretation and validation in larger cohorts (Figure 6).
Using corrected odds ratios derived from contingency tables, notable associations emerged. The presence of disseminated lesions and prolonged symptom duration (>4 weeks) were each linked to an approximately 11-fold increase in the likelihood of unfavorable clinical outcomes. In addition, high-grade histological subtypes—specifically NUT carcinoma and squamous cell carcinoma—were associated with a markedly elevated risk (corrected OR ≈ 55). These effect size estimates should be interpreted as exploratory signals requiring cautious follow-up.
- Additionally, in theDiscussion section (lines 410–413), a comment has been introduced highlighting delayed diagnosis as a key contributing factor to adverse clinical outcomes.
In our cohort, prolonged symptom duration was associated with an approximately 11-fold increase in the likelihood of unfavorable clinical outcome (corrected OR ≈ 11.0), reinforcing the detrimental impact of delayed diagnosis even at the individual case level.
- Finally, a newly developed paragraph within theLimitations section (lines 541-544) now addresses the restricted follow-up duration and its impact on long-term survival estimations.
Moreover, the brevity of clinical follow-up hindered a meaningful assessment of long-term survival, as the available data lacked sufficient temporal depth to support reliable estimates beyond early outcome horizons.
These revisions are contextually integrated throughout the manuscript and contribute to a more coherent and methodologically transparent presentation.
- Parts of the Results (e.g., bronchoscopic and histopathological descriptions) are rich in detail but contain some repetition. Would the authors consider restructuring or summarizing key diagnostic and therapeutic findings more concisely—perhaps with an additional table?
Response:
Alongside Table 1, which already summarizes tumor types, demographic data, and clinical presentation, a new table (Table 3) has now been added to further enhance the clarity of the results. This table presents details on the genetic mutations and rearrangements investigated in individual patients, the diagnostic modalities used to detect them, and targeted therapies aimed at specific signaling pathways that were applicable within our institutional setting. We believe this addition has improved the readability and completeness of the dataset. Consequently, the previously numbered Table 3 is now designated as Table 4. The new table is located in lines 332–341 of the revised manuscript.
Additionally, in response to your thoughtful observation, parts of the Materials and Methods and Results sections have been restructured to reduce redundancy and improve clarity. Several repetitive sentences were removed, and minor stylistic and linguistic edits were applied throughout the section. These revisions have contributed to a more concise and focused presentation of diagnostic and therapeutic findings. All modifications are visible in the tracked changes, spanning multiple subsections within Materials and Methods and Results.
- The paper emphasizes the value of genetic testing and targeted therapies (e.g., crizotinib in IMT with TFG::ROS1 fusion), but it lacks discussion of their availability and cost in resource-limited settings—the very context the paper claims to address. Could the authors expand on these practical considerations?
Response:
We appreciate the reviewer’s important observation. We have added a dedicated paragraph to the Discussion section (lines 477–493) that directly addresses the challenges of accessing genetic testing and targeted therapies in resource-limited settings. This passage outlines the availability and cost barriers in LMICs and highlights the role of international collaborations and context-sensitive treatment strategies. Furthermore, an additional paragraph (lines 527–538) explores practical therapeutic dilemmas relevant to crizotinib use in IMT, including questions regarding timing, duration, resistance, and access to molecular diagnostics. Together, these additions aim to ground the paper’s translational relevance in the realities of constrained healthcare environments.
- The summary and conclusions seem to offer prescriptive strategies, but the data are essentially from an uncontrolled series of cases. Would the authors consider clarifying that these are observations from a case series rather than formal, validated recommendations?
Response:
We thank the reviewer for this insightful comment. We fully acknowledge that the data derive from an uncontrolled case series and, as such, do not support formal or universally generalizable recommendations. In response, we have clarified in both the Discussion and Conclusion sections that the proposed diagnostic and therapeutic considerations should be viewed as observational insights rather than prescriptive guidelines. Furthermore, the Limitations paragraph (lines 539–550) explicitly states that the small sample size, retrospective design, and single-center nature of the study restrict the extrapolation of our findings. These revisions were made to ensure that the conclusions are interpreted within the proper clinical and methodological context.
Round 2
Reviewer 3 Report
Comments and Suggestions for Authors
The authors have complied with most of the reviewer's comments. They corrected the text of the article and thus contributed to its substantive value.